# LVQ-VAE: End-to-end Hyperprior-based Variational Image Compression with Lattice Vector Quantization

## Abstract

Image compression technology has become more important research topic. In recent years, learning-based methods have been extensively studied and variational autoencoder (VAE)-based methods using hyperprior-based context-adaptive entropy model have been reported to be comparable to the latest video coding standard H.266/VVC in terms of RD performance. We think there is room for improvement in quantization process of latent features by adopting vector quantization (VQ). Many VAE-based methods use scalar quantization for latent features and do not exploit correlation between the features. Although there are methods that incorporate VQ into learning-based methods, to the best our knowledge, there are no studies that utilizes the hyperprior-based VAE with VQ because incorporating VQ into a hyperprior-based VAE makes it difficult to estimate the likelihood. In this paper, we propose a new VAE-based image compression method using VQ based latent representation for hyperprior-based context-adaptive entropy model to improve the coding efficiency. The proposed method resolves problem faced by conventional VQ-based methods due to codebook size bloat by adopting Lattice VQ as the basis quantization method and achieves end-to-end optimization with hyperprior-based context-adaptive entropy model by approximating the likelihood calculation of latent feature vectors with high accuracy using Monte Carlo integration. Furthermore, in likelihood estimation, we model each latent feature vector with multivariate normal distribution including covariance matrix parameters, which improves the likelihood estimation accuracy and RD performance. Experimental results show that the proposed method achieves a state-of-the-art RD performance exceeding existing learning-based methods and the latest video coding standard H.266/VVC by 18.0 % for Kodak, 21.9 % for CLIC2022 and 39.2 % for Tecnick.

## 1 Introduction

Image compression technology has become more important than ever to achieve efficient data transmission and storage due to the demand for of high-quality contents and the increase in the popularity of video services. Various conventional image compression technologies have been standardized so far (JPEG (Wallace, 1991; ITU, 1993), JPEG2000 (Taubman & Marcellin, 2002; ISO/TEC, 2004), WebP (Google), H.264/AVC (Marpe et al., 2006; ISO/IEC, 2003), H.265/HEVC (Sullivan et al., 2012; ISO/IEC, 2013), H.266/VVC (Bross et al., 2021; ISO/IEC, 2020), etc.). These technologies consist of a combination of transform, quantization and entropy coding. Transform is a major part of JPEG, H.265/HEVC and H.266/VVC which use DCT or DST, while JPEG2000 uses wavelet transform, all of which are based on handcrafted linear transforms. These hand-crafted design are limited in their ability to capture features for a variety of images.

In recent years, deep learning has made remarkable progress, and learning-based methods are being actively explored in the field of image compression. Most recent learning-based methods are based on transform coding (Goyal, 2001). Many of these methods use convolutional neural network (CNN)-based autoencoders in which the encoder transforms the input image into a latent representation and then performs quantization and entropy coding, while the decoder reconstructs the restored image. This approach achieves flexible nonlinear transforms that have higher potential to

map pixels into a more compressible latent representation than the linear transforms used by classical image compression approaches. It can be divided into two types according to the metric used for encoder optimization. One is the generative approach that directly maximizes subjective image quality (Rippel & Bourdev, 2017; Santurkar et al., 2018; Agustsson et al., 2019; Mentzer et al., 2020; Kudo et al., 2021). This approach aims to optimize the distribution of reconstructed images to approach that of natural images by using generative adversarial networks. The other maximizes an objective metric such as peak signal-to-noise ratio (PSNR). This approach solves the rate-distortion (RD) optimization problem in the same way as the classical image compression described above. This paper focuses on the latter approach as it is applicable to a wider range of applications. The latter approach is found in various proposals. Toderici et al. (2016; 2017) introduced recurrent neural networks for feature extraction and Johnston et al. (2017) enhanced these networks to improve the coding performance. Cai & Zhang (2018); Cai et al. (2018) directly trained the quantization. These methods quantize the latent features as fixed-length codes.

By contrast, variational autoencoder (VAE)-based methods have been proposed that formulate the optimization as the problem of minimizing the entropy of the quantized latent features as well as the expected distortion of the reconstructed image with respect to the original. The first image compression method using VAE was proposed by Theis et al. (2017) and Ballé et al. (2017). They studied entropy models to approximate the actual distributions of the quantized latent features. To improve the accuracy of the entropy model, a hyperprior-based context-adaptive entropy model was proposed by Ballé et al. (2018); it has been the baseline in most subsequent research. Whereas the actual distributions of each latent feature are fixed in (Theis et al., 2017; Ballé et al., 2017), Ballé et al. (2018) approximated the entropy model as a zero-mean Gaussian distribution with scale parameter for each latent feature to remove the spatial redundancy, where contexts are encoded as side information. Based on this hyperprior-based context-adaptive entropy model, various methods have been proposed to estimate the entropy model with higher accuracy. The autoregressive context model is one of the technologies that has experienced significant performance improvements. Minnen et al. (2018) and Lee et al. (2019) proposed to jointly utilize an autoregressive context model and the mean and scale hyperpriors. Mentzer et al. (2018) and Chen et al. (2021) extended an autoregressive context model that utilizes channel neighbors with 3D Masked Convolution module. In (Minnen & Singh, 2020) and (Zhu et al., 2022b), the channel-directed autoregressive model was applied to reduce the computational complexity of the spatial-directed autoregressive model and He et al. (2022) was further improved by dividing the model non-evenly into channel directions. To further improve the entropy model, Liu et al. (2020) and Cheng et al. (2020) proposed a Gaussian mixture model and developed a network architecture by adopting an attention module. As another improvement perspective, Hu et al. (2020) proposed coarse-to-fine hyperprior modeling while Yang et al. (2020) improved the performance by devising an inference process without changing the training process. Ho et al. (2021) and Xie et al. (2021) focused on improving the network architecture by adopting a normalizing flow module. Some methods have been reported to better the RD performance of H.266/VVC, the latest video coding standard, which is not learning based, in terms of the MS-SSIM metric, but are still comparable in the PSNR metric.

Vector quantization (VQ) is incorporated into learning-based methods to leverage performance. Since VQ potentially offers better performance than scalar quantization in terms of RD (Gray & Neuhoff, 1998; Gray, 1984; Chou et al., 1989), various studies have examined it (Shin & Lu, 1991; Antonini et al., 1992; Tatsaki et al., 1995; Shnaider & Paplinski, 2001; Voinson et al., 2002; Salleh & Soraghan, 2007; Chiranjeevi & Jena, 2018; Nag, 2019). The challenge in applying VQ to learning-based compression methods is how to incorporate the likelihood estimation of latent feature vectors into the optimization process. van den Oord et al. (2017) proposed VQ-VAE which avoids likelihood calculations by assuming uniformity of the prior distribution of latent features and separately designing encoder/decoder and codebooks to perform gradient optimization. Razavi et al. (2019) and Fauw et al. (2019) extended VQ-VAE to a hierarchical network structure. Williams et al. (2020) revised the quantization process while Xue et al. (2019) combined optimization with supervised learning. However, all of the above methods do not perform well because the learning parameters underlying the encoder/decoder and codebook are designed separately to achieve gradient optimization and/or the probability distribution is assumed to be uniform. Agustsson et al. (2017) attempts end-to-end optimization by performing VQ using a soft-to-hard annealing strategy. However, its learning suffers from unstable convergence, because it approximates the prior distribution of the codebook with a histogram taken from the training process. Zhu et al. (2022a) proposed a cascaded vector quantization with multi-codebooks to keep memory

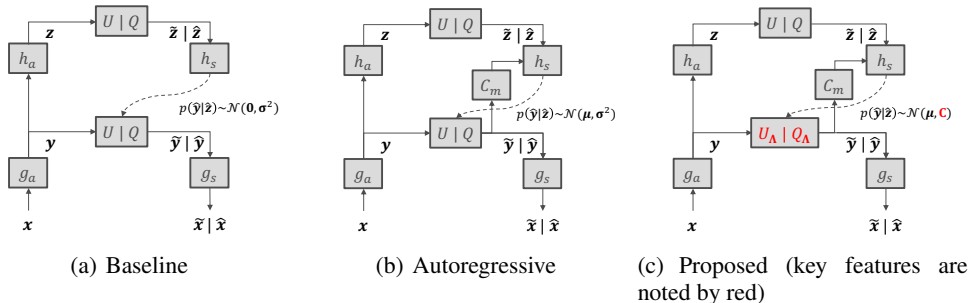

Figure 1: Operational diagrams of VAE-based image compression. $U \mid Q$ and $U_{\boldsymbol{\Lambda}} \mid Q_{\boldsymbol{\Lambda}}$ represents either quantization during training (producing variables with a tilde) or quantization during inference (producing variables with a hat).

capacity and performance at high bitrates. However, due to the limitation on the vector dimension, this approach does not fully utilize the potential performance of vector quantization. In addition, to the best knowledge of the authors, no study has utilized the above-mentioned hyperprior-based context-adaptive entropy model with VQ. Incorporating VQ into the hyperprior-based method and optimizing them in an end-to-end manner has not been achieved, because incorporating VQ into a hyperprior-based context-adaptive entropy model makes it difficult to estimate the likelihood.

In this paper, we propose a new VAE-based image compression method using VQ-based latent representation for a hyperprior-based context-adaptive entropy model (named LVQ-VAE) with improved coding efficiency. The proposed method has two main features. First we introduce Lattice VQ for quantization, which resolves the codebook size bloat problem faced by conventional VQ-based methods without coding efficiency degradation. At the same time, the proposed method achieves end-to-end optimization with the hyperprior-based context-adaptive entropy model by using Monte Carlo integration to approximate with high accuracy the likelihood calculation of latent feature vectors. Second, in likelihood estimation, we model each latent feature vector as a multivariate normal distribution including covariance matrix parameters, which improves the likelihood estimation accuracy and RD performance. Zhu et al. (2022a) also models latent feature vectors as a multivariate normal distribution, but this method models the features contained in each representative point as a single distribution, whereas the proposed method models each feature vector, which eliminates correlation among feature vectors and is expected to improve coding performance.

The contributions of this paper are summarized as follows:

- We incorporate VQ into VAE-based image compression method using hyperprior-based context-adaptive entropy model while optimizing them for end-to-end manner, which leads to obtain better RD performance that exceeds the limits possible with scalar quantization without codebook size bloat.

- We improve the entropy model by modeling each latent feature vector as a multivariate normal distribution including covariance matrix parameters, which improves RD performance.

- We show that the proposed method achieves state-of-the-art RD performance compared to existing learning-based methods and the latest video coding standard H.266/VVC with regard to the PSNR metric.

## 2 VAE-BASED IMAGE COMPRESSION

Most recent VAE-based methods are based on the hyperprior model developed by Ballé et al. (2018) as shown in Fig. 1 (a). This model consists of four parametric transforms:

- $g_a(\boldsymbol{x}; \boldsymbol{\phi}_g)$: a feature encoder that transforms input image $\boldsymbol{x}$ into latent feature $\boldsymbol{y}$, which is quantized to form $\hat{\boldsymbol{y}} = Q(\boldsymbol{y})$,

- $g_s(\hat{\boldsymbol{y}}; \boldsymbol{\theta}_g)$: a feature decoder that reconstructs image $\hat{\boldsymbol{x}} = g_s(\hat{\boldsymbol{y}}; \boldsymbol{\theta}_g)$,

- $h_a(\boldsymbol{y}; \boldsymbol{\phi}_h)$: a hyper-encoder that extracts latent representation $\boldsymbol{z}$ for context information, which is quantized to form $\hat{\boldsymbol{z}} = Q(\boldsymbol{z})$,

- $h_s(\hat{\boldsymbol{z}}; \boldsymbol{\theta}_h)$: a hyper-decoder that generates the context information for estimating the entropy model $p_{\hat{\boldsymbol{y}}|\hat{\boldsymbol{z}}}(\hat{\boldsymbol{y}} \mid \hat{\boldsymbol{z}})$.

$\boldsymbol{\phi}_g, \boldsymbol{\theta}_g, \boldsymbol{\phi}_h, \boldsymbol{\theta}_h$ are optimized parameters of each transform, which are generally composed of neural networks such as CNNs. $U \mid Q$ denotes a quantizer in a training phase ($U$) and one in an inference phase ($Q$). In the training phase, quantizer $U$ is approximated using additive uniform noise as $\tilde{\boldsymbol{y}} = U(\boldsymbol{y}) = [y_1 + u_1, \ldots, y_N + u_N]$, where $u_i$ is sampled from univariate uniform distribution $\mathcal{U}(-\frac{1}{2}, \frac{1}{2})$. This is because end-to-end learning requires the quantization to realize gradient-based optimization. In addition to the above approximation, several other approximation techniques have been studied, such as stochastic binarization (Toderici et al., 2016), universal quantization (Choi et al., 2019), straight-through estimator (van den Oord et al., 2017), and soft-to-hard quantization (Agustsson et al., 2017). In the inference phase, quantizer $Q$ is actual quantization such as a rounding operator. In this manuscript, we represent approximated data as a variable with a tilde and quantized data as one with a hat as shown in Fig. 1.

The encoder compresses latent representation $\hat{\boldsymbol{y}}$ and $\hat{\boldsymbol{z}}$ by using entropy coding such as arithmetic coding (Rissanen & Langdon, 1979) and transmits it as a bitstream. The entropy coding estimates probability $p_{\hat{\boldsymbol{y}}|\hat{\boldsymbol{z}}}(\hat{\boldsymbol{y}} \mid \hat{\boldsymbol{z}})$ as zero-mean Gaussian $\mathcal{N}(\boldsymbol{0}, \boldsymbol{\sigma}^2)$; its context information is scale parameter $\boldsymbol{\sigma} = h_s(\hat{\boldsymbol{z}}; \boldsymbol{\theta}_h)$. To further improve this estimation, Minnen et al. (2018) and Lee et al. (2019) jointly utilized the autoregressive model and the mean and scale parameters of Gaussian distribution (Fig. 1 (b)), where $C_m$ denotes a context prediction model conditioned on decoded features that are composed of an autoregressive module such as masked convolutions (van den Oord et al., 2016). As for $p_{\hat{\boldsymbol{z}}}(\hat{\boldsymbol{z}})$, the non-adaptive fixed density model called factorized prior is trained and shared between the encoder and the decoder.

Finally, the learning process is formulated as RD optimization which minimizes the following loss function.

$$
\begin{aligned}
\mathcal{L} &= \mathcal{R} + \lambda \cdot \mathcal{D} \\
&= \mathbb{E}_{\boldsymbol{x} \sim p_{\boldsymbol{x}}} \left[ -\log p_{\tilde{\boldsymbol{y}}|\tilde{\boldsymbol{z}}}(\tilde{\boldsymbol{y}} \mid \tilde{\boldsymbol{z}}) - \log p_{\tilde{\boldsymbol{z}}}(\tilde{\boldsymbol{z}}) \right] + \lambda \cdot \mathbb{E}_{\boldsymbol{x} \sim p_{\boldsymbol{x}}} \left\| \boldsymbol{x} - \tilde{\boldsymbol{x}} \right\|_2^2
\end{aligned}
\tag{1}
$$

where $p_{\boldsymbol{x}}$ is the marginal distribution of natural images, and $\mathcal{D}$ represents the expected distortion between the reconstructed image and the original, and $\mathcal{R}$ represents the entropy of $\tilde{\boldsymbol{y}}$ and $\tilde{\boldsymbol{z}}$ that approximates the expected code length of the bitstream. $\lambda$ is the Lagrange multiplier that controls the RD trade-off.

## 3 PROPOSED METHOD

The proposed method has two key features:

1. We introduce Lattice VQ for quantization of the latent features $\boldsymbol{y}$, as this achieves better RD performance than scalar quantization without codebook size bloat.

2. We model each latent feature vector with a multivariate normal distribution including covariance matrix parameters and incorporate it into the autoregressive model to achieve highly accurate likelihood estimation.

Figure 1 (c) shows the operational diagram of the proposed method. In the following, we will describe a quantizer design based on Lattice VQ and then explain how to incorporate it into a hyperprior-based context-adaptive entropy model.

### 3.1 QUANTIZER DESIGN BASED ON LATTICE VECTOR QUANTIZATION

We use Lattice VQ for quantizing latent feature $\boldsymbol{y}$. Given $N$ elements of features denoted by $y_1, \ldots, y_N$, the features are split into $n$-dimensional vectors $\boldsymbol{v}_1, \ldots, \boldsymbol{v}_{\lceil N/n \rceil}$ as follows.

$$
\boldsymbol{v}_i = [y_{ni-n+1}, y_{ni-n+2}, \ldots, y_{ni}], 1 \leq i \leq \lceil N/n \rceil
\tag{2}
$$

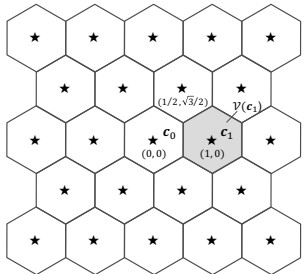

Figure 2: $A_2$ lattice

Then these $n$-dimensional vectors are divided into clusters, and each cluster is represented by its centroid point.

Lattice VQ is a VQ that places representative points at the location to form a lattice. Lattice $\mathbf{\Lambda} \in \mathbb{R}^n$ is formed as a linear combination of basis vectors (Conway & Sloane, 1988):

$$\mathbf{\Lambda} = \left\{ \sum_{i}^{n} k_i \boldsymbol{b}_i \mid k_i \in \mathbb{Z} \right\} \tag{3}$$

where $\boldsymbol{b}_1, \ldots, \boldsymbol{b}_n$ are basis vectors for $\mathbb{R}^n$, and a matrix with these basis vectors as column vectors is also called the generating matrix which is uniquely defined for the structure of lattice, and $k_i$ is an integer coefficient. For example, the 2-dimensional hexagonal lattice $A_2$ is shown in Fig. 2. The lattice points $\boldsymbol{c} \in \mathbf{\Lambda}$ are the representative points of Lattice VQ. Let $Q_{\mathbf{\Lambda}}(\boldsymbol{v})$ be the nearest neighbor of $\boldsymbol{v}$ in terms of Euclidean norm in the lattice:

$$Q_{\mathbf{\Lambda}}(\boldsymbol{v}) = \{\boldsymbol{c}_i : \|\boldsymbol{v} - \boldsymbol{c}_i\| \leq \|\boldsymbol{v} - \boldsymbol{c}_j\|, \boldsymbol{c}_i, \boldsymbol{c}_j \in \mathbf{\Lambda} \text{ for all } j \neq i\} \tag{4}$$

The Voronoi region of lattice point $\boldsymbol{c}_i$ is the set of all vectors mapped into this point as defined by

$$\mathcal{V}(\boldsymbol{c}_i) = \{\boldsymbol{v} : \|\boldsymbol{v} - \boldsymbol{c}_i\| \leq \|\boldsymbol{v} - \boldsymbol{c}_j\|, \boldsymbol{c}_i, \boldsymbol{c}_j \in \mathbf{\Lambda} \text{ for all } j \neq i\} \tag{5}$$

Key features of Lattice VQ are follows:

1. A fast quantization method has been developed. Several lattices ($A_2$, $D_4$ (4-dimensional checkerboard root lattice), $E_8$ (8-dimensional Gosset's root lattice), $\Lambda_{24}$ (24-dimensional Leech lattice), etc.) have fast quantization methods (Conway & Sloane, 1982; 1986; Vardy & Be'ery, 1993), and there are also fast iterative methods such as (Agrell et al., 2002) for arbitrary lattices.

2. Lattice VQ with a particular generator matrix such as $A_2$, $D_4$, $E_8$, $\Lambda_{24}$ is an optimal quantizer minimizing the distortion for uniform distribution source.

In terms of effectively utilizing the potential of Lattice VQ, we set the vectorization axis in the channel direction. There are two reasons for this. One is that the autoregressive model based on a pixel-by-pixel update cannot be applied when the vectorization axis is set in the spatial direction. The other is that most conventional VAE-based methods do not take account of correlations in the channel direction. Although Minnen & Singh (2020); Zhu et al. (2022b); He et al. (2022) apply autoregressive models in the channel direction, the distribution of latent features is indirectly predicted using features obtained by AR models. By contrast, the proposed method predicts the distribution of latent features directly, which is expected to improve performance.

To perform end-to-end optimization with the gradient method, we employ the approximation method proposed by Lee et al. (2019), that is, we use straight-through estimator (STE) (Courbariaux & Bengio, 2016) for a decoder and the additive noise (Ballé et al., 2018) for the entropy model. Unlike when using additive noise for both as is common in most conventional methods, this method can eliminate train-test mismatch. Let $\mathcal{U}_{\mathcal{V}_0}$ be a $n$-dimensional random variable uniformly distributed over the basic cell of the lattice $\mathcal{V}_0 = \mathcal{V}(\mathbf{0})$, the Voronoi region of the lattice

point $\mathbf{0}$. We generate $\boldsymbol{u}_i \sim \mathcal{U}_{\mathcal{V}_0}$ and perform the following process.

$$\tilde{\boldsymbol{v}}_i = U_{\boldsymbol{\Lambda}}(\boldsymbol{v}_i) = \begin{cases} Q_{\boldsymbol{\Lambda}}^{\text{STE}}(\boldsymbol{v}_i) & \text{for a decoder} \\ \boldsymbol{v}_i + \boldsymbol{u}_i & \text{for an entropy model} \end{cases} \qquad \text{(in training)} \qquad (6)$$

$$\hat{\boldsymbol{v}}_i = Q_{\boldsymbol{\Lambda}}(\boldsymbol{v}_i) \qquad \text{(in inference)} \qquad (7)$$

where $Q_{\boldsymbol{\Lambda}}^{\text{STE}}$ is the same as $Q_{\boldsymbol{\Lambda}}$ but its gradient is $\frac{\partial}{\partial \boldsymbol{v}_i} \tilde{\boldsymbol{v}}_i = \mathbf{1}$.

## 3.2 PROBABILITY MODEL

The probability entropy model for $\hat{\boldsymbol{v}}$ is represented as

$$p_{\hat{\boldsymbol{v}}|\hat{\boldsymbol{z}}}(\hat{\boldsymbol{v}} \mid \hat{\boldsymbol{z}}) = \prod_{i=1}^{\lceil N/n \rceil} p_{\hat{\boldsymbol{v}}_i|\hat{\boldsymbol{z}}}(\hat{\boldsymbol{v}}_i \mid \hat{\boldsymbol{v}}_{<i}, \hat{\boldsymbol{z}}) \qquad (8)$$

where $\hat{\boldsymbol{v}}_{<i} = [\hat{\boldsymbol{v}}_1, \hat{\boldsymbol{v}}_2, \ldots, \hat{\boldsymbol{v}}_{i-1}]$. Each element of vectorized features $\hat{\boldsymbol{v}}_i$ is modeled as a multivariate Gaussian with its own mean $\boldsymbol{\mu}_i$ and covariance matrix $\mathbf{C}_i$, where the mean and covariance matrix are jointly predicted by the hyper-decoder $h_s(\hat{\boldsymbol{z}}; \boldsymbol{\theta}_h)$ and autoregressive neural networks $f_\psi$ consisting of masked convolution:

$$p_{\hat{\boldsymbol{v}}_i|\hat{\boldsymbol{z}}}(\hat{\boldsymbol{v}}_i \mid \hat{\boldsymbol{v}}_{<i}, \hat{\boldsymbol{z}}) = \int_{\hat{\boldsymbol{v}}_i + \mathcal{V}_0} \mathcal{N}(\boldsymbol{\mu}_i, \mathbf{C}_i)(\boldsymbol{x}) \, d\boldsymbol{x} \qquad (9)$$

where $\boldsymbol{\mu}_i, \mathbf{C}_i$ are estimated using $f_\psi(\hat{\boldsymbol{v}}_{<i}, h_s(\hat{\boldsymbol{z}}; \boldsymbol{\theta}_h))$. As for the probability entropy model of $\hat{\boldsymbol{z}}$, we use the factorized density model that is the same method as used in the previous work (Ballé et al., 2018).

### 3.2.1 COVARIANCE MATRIX ESTIMATION

It is necessary to estimate covariance matrix $\mathbf{C}_i$ to satisfy the condition of a positive definite symmetric matrix. It is known that the positive definite symmetric matrix can be generated by matrix operation on an arbitrary matrix and its transpose. Here, network $f_\psi(\hat{\boldsymbol{v}}_{<i}, h_s(\hat{\boldsymbol{z}}; \boldsymbol{\theta}_h))$ outputs mean vector $\boldsymbol{\mu}_i \in \mathbb{R}^n$ and matrix $\mathbf{A}_i \in \mathbb{R}^{n \times n}$ for each vectorized feature:

$$f_\psi(\hat{\boldsymbol{v}}_{<i}, h_s(\hat{\boldsymbol{z}}; \boldsymbol{\theta}_h)) = \begin{bmatrix} \boldsymbol{\mu}_i \\ \mathbf{A}_i \end{bmatrix} \qquad (10)$$

$$\boldsymbol{\mu}_i = [\mu_1, \quad \mu_2, \quad \cdots, \quad \mu_n] \qquad (11)$$

$$\mathbf{A}_i = \begin{bmatrix} a_{1,1}, & a_{1,2}, & \cdots, & a_{1,n} \\ a_{2,1}, & a_{2,2}, & \cdots, & a_{2,n} \\ \vdots & \vdots & \ddots & \vdots \\ a_{n,1}, & a_{n,2}, & \cdots, & a_{n,n} \end{bmatrix} \qquad (12)$$

Then, covariance matrix $\mathbf{C}_i$ is calculated as

$$\mathbf{C}_i = \frac{1}{n} \mathbf{A}_i^T \mathbf{A}_i + \varepsilon \mathbf{I}, \qquad (13)$$

where the second term stabilizes the estimation and $\varepsilon$ is a small value. We set $\varepsilon$ to $10^{-2}$ in our experiment.

### 3.2.2 PROBABILITY CALCULATION

It is numerically difficult to compute the integration of Eq. (9) because the integration region $\hat{\boldsymbol{v}}_i + \mathcal{V}_0$ is complex polytope in most lattices. Furthermore, with increasing dimensionality, the integral computations become impractical in terms of computational complexity. To resolve this, we introduce a Monte Carlo (MC) method (MacKay, 1998), which approximates integration as

$$\int_{\hat{\boldsymbol{v}}_i + \mathcal{V}_0} \mathcal{N}(\boldsymbol{\mu}_i, \mathbf{C}_i)(\boldsymbol{x}) \, d\boldsymbol{x} \approx \frac{1}{M} \sum_{j=1}^{M} \frac{\mathcal{N}(\boldsymbol{\mu}_i, \mathbf{C}_i)(\boldsymbol{x}_j)}{p(\boldsymbol{x}_j)}, \qquad (14)$$

where $\boldsymbol{x}_j$ is an $n$-dimensional point sampled from arbitrary probability density function $p$. The method also has the advantage of being applicable to more complex probability distributions, since end-to-end optimization is possible as long as the probability density function and its derivative can be computed.

To further improve estimation accuracy, we adopt a Quasi-Monte Carlo (QMC) method (Radovic et al., 1996), which uses low-discrepancy sequences such as Halton, Sobol', and Faure sequences as sampling points. The advantage of the QMC method is a faster rate of convergence than the naive MC method. The QMC method has a rate of convergence close to $O(1/M)$, whereas the rate for the MC method is $O(1/\sqrt{M})$. Thus Eq. (14) is re-written as

$$\int_{\hat{\boldsymbol{v}}_i + \mathcal{V}_\mathbf{0}} \mathcal{N}(\boldsymbol{\mu}_i, \mathbf{C}_i)(\boldsymbol{x})\,d\boldsymbol{x} \approx \frac{V}{M} \sum_{j=1}^{M} \mathcal{N}(\boldsymbol{\mu}_i, \mathbf{C}_i)(\boldsymbol{x}_j) \tag{15}$$

where $V = \frac{1}{p(\boldsymbol{x}_j)}$ represents the volume of the Voronoi region. In the experiment section, we employ Sobol's sequence as sampling points and set the number of sampling points, $M$, to 256 in training and 8192 in inference.

Since the MC/QMC method contains estimation error, the following two processes are introduced during inference to prevent decoding failure. First, when calculating the cumulative probabilities in both the encoder and the decoder, the cumulative probabilities are accumulated in order of the centroid closest to each estimated mean vector, as long as the cumulative value does not exceed 1. The feature vector is quantized to the nearest centroid with the smallest distance among calculated centroids. Second, the encoder and the decoder use the fixed samples, namely, they use the same (fixed) random seed when calculating the probability with the QMC method in order to match the probability values of the encoder and the decoder.

## 4 EXPERIMENTS

We conducted simulations to verify the efficiency of the proposed method.

### 4.1 NETWORK ARCHITECTURE

Our network architecture is based on (Cheng et al., 2020) (See Appendix A.1 for details). $N$ is channel size corresponding to the network capacity, which is set according to the bitrate as described below. As we use multivariate Gaussian model, the output of $f_\psi$ requires $N(n+1)$ channels.

### 4.2 EXPERIMENT CONDITIONS

We used $\Lambda_{24}$ lattice ($n = 24$). The reason is that $\Lambda_{24}$ gives the optimal representation for a uniform distribution. Hence it is expected that the proposed method brings its latent features towards a uniform distribution through a optimization process ($A_2, D_4, E_8$ in Sec. 4.3.1 are used for the same reason). We adopted a fast quantization method for LVQ, which does not need to store representative points and offers a low complexity as described in Appendix A.2. We used a subset of ImageNet dataset (Russakovsky et al., 2015), and randomly scaled and cropped them to the size of $192 \times 192$ during training. We used Adam optimizer(Kingma & Ba, 2015) with a batch size of 16. The learning rate was scheduled at $10^{-4}$ for the first 20 epochs, and reduced to $10^{-5}$ for the last 10 epochs. $\lambda$ was set to $\{256, 1024, 2048, 4096\}$ and network channel size $N$ was set to 144 for the lowest rate model and 192 for three higher rate models. For evaluation, we tested the Kodak image dataset (Kodak, 1993) consisting of 24 images (See Appendix B for other datasets). To evaluate the RD performance, the bitrate is measured by bits per pixel (bpp), and the quality is measured by PSNR, where bitrate and PSNR are calculated by averaging the encoding results for all 24 images. We plot the RD curves and calculate the Bjontegaard delta bitrate (BDBR) (Bjontegaard, 2001) to compare their coding efficiency. We compared the proposed method with VTM 15.0 (JVET), the reference software of the latest video coding standard H.266/VVC, with intra profile and H.265/HEVC-based encoder BPG (Bellard) and several of the learning-based methods. For VTM and BPG, the input RGB images were converted into YUV444 format and encoded, then the reconstructed images were converted into RGB and PSNR were calculated by averaging the MSE of each RGB value. For the

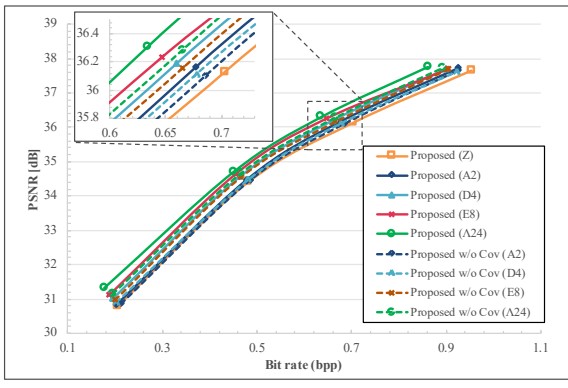

Figure 3: Comparison of RD performance for Kodak dataset for different number of dimensions and the use or non-use of a covariance matrix in the proposed method

Table 1: Comparison of BDBR for Kodak dataset for different numbers of dimensions and the use or non-use of a covariance matrix in the proposed method. $A_2, D_4, E_8, \Lambda_{24}$ are the optimal lattice quantizers for 2-, 4-, 8- and 24-dimension as mentioned Sec. 3.1.

| Dimension $n$ | C | BDBR |
|---|---|---|
| 1 ($\mathbb{Z}$, scalar) | - | 0.0 % (anchor) |
| 2 ($A_2$) | - | 0.0 % |
| | ✓ | -2.3 % |
| 4 ($D_4$) | - | -1.8 % |
| | ✓ | -6.8 % |
| 8 ($E_8$) | - | -6.3 % |
| | ✓ | -10.0 % |
| 24 ($\Lambda_{24}$) | - | -9.2 % |
| | ✓ | **-14.7 %** |

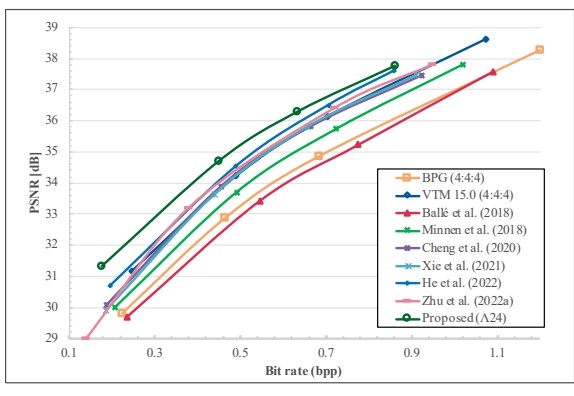

Figure 4: RD performance for Kodak dataset

Table 2: Comparison of BDBR for Kodak dataset

| Method | BDBR |
|---|---|
| BPG (4:4:4) | 23.8 % |
| VTM 15.0 (4:4:4) | 0.0 % (anchor) |
| Ballé et al. (2018) | 29.8 % |
| Minnen et al. (2018) | 11.4 % |
| Cheng et al. (2020) | 0.5 % |
| Xie et al. (2021) | 1.4 % |
| He et al. (2022) | -6.4 % |
| Zhu et al. (2022a) | -2.7 % |
| Proposed ($\Lambda_{24}$) | **-18.0 %** |

learning-based methods, including the proposed method, we used RGB format as the input. For the proposed method, the bitrate is calculated by the estimated values ($-\log_2(p)$).

## 4.3 RESULTS AND DISCUSSIONS

### 4.3.1 ABLATION STUDY

In order to clarify the effect of the number of dimensions $n$ of Lattice VQ and the use of the estimated covariance matrix in the proposed method, we additionally simulate in other dimensions ($n = 1$ ($\mathbb{Z}$, scalar quantization), $n = 2(A_2)$, $n = 4(D_4)$, $n = 8(E_8)$) and compared the performance with and without the estimation of the covariance matrix. Without estimation of the covariance matrix, $f_\psi$ output $2N$ channels ($n$ means and $n$ scales for each feature vector) and squared scales were aligned to the diagonal components of each matrix $\mathbf{C}_i$; the other elements were set to 0.

The results of RD curves and BDBR are shown in Fig. 3 and Tab. 1. As for the number of dimensions, it can be shown that RD performance increases with the number of dimensions. This is consistent with the property of VQ that the expected error decreases as the number of dimensions increases, and there is the potential for further increases in gain with dimensionality increases. Regarding the use of a covariance matrix, the results shows better performance when a covariance matrix is used. This suggests the presence of correlation within the feature vector, which could be eliminated by using a covariance matrix as mentioned Sec. 3.1.

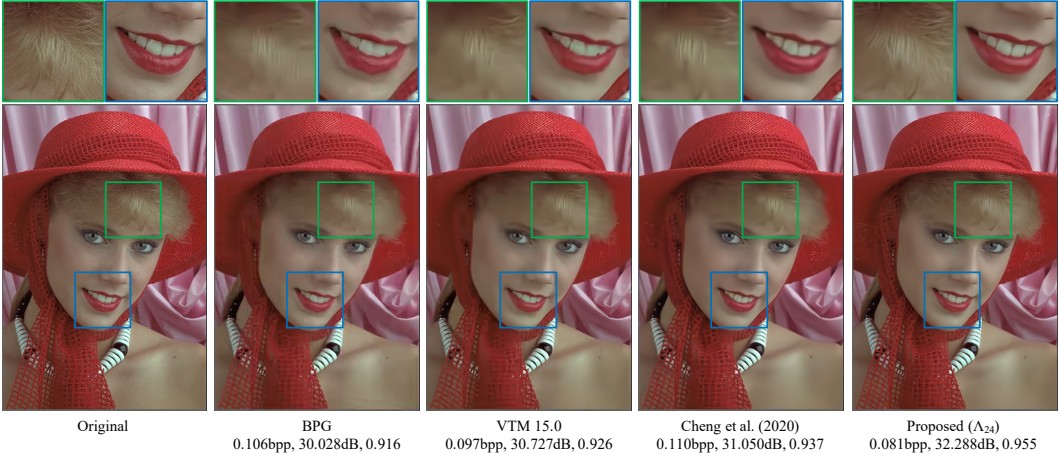

Figure 5: Reconstructed images of kodim04 (bpp, PSNR, MS-SSIM)

### 4.3.2 RD PERFORMANCE

Fig. 4 and Tab. 2 show the results of comparison between the proposed method and existing methods in terms of RD performance. The results show that the proposed method gave better RD performance than all previous learning-based methods. Furthermore, the proposed method outperforms VTM 15.0 by 18.0 %.

### 4.3.3 QUALITATIVE EVALUATION

To evaluate the qualitative performance, we visualized the reconstructed images. Fig.5 shows reconstructed images at the level of approximately 0.1 bpp. In Fig. 5, the proposed method maintains more detail, such as the woman's hair and the contours of teeth, than the other methods. Moreover, it is observed that the other methods suffer from some artifacts and degradation.

## 5 CONCLUSION

This paper proposed a new VAE-based image compression method characterized by Lattice VQ for improving the hyperprior-based context-adaptive entropy model approach. The proposed method achieves end-to-end optimization with a hyperprior-based context-adaptive entropy model by approximating the likelihood calculation of latent feature vectors with high accuracy by using Monte Carlo integration. Furthermore, the proposed method provides highly accurate likelihood estimation by modeling the distribution parameters of latent feature vectors.

Experiments on public data sets showed that the proposed method achieves state-of-the-art RD performance compared to existing learning-based methods and outperforms VTM 15.0, the reference software of the latest video coding standard H.266/VVC, by 18.0 % for Kodak, 21.9 % for CLIC2022 and 39.2 % for Tecnick in the PSNR metric.

As a future work, we will address to reduce the complexity. This paper pursue maximizing coding efficiency rather than reducing complexity. Therefore, compared to the latest methods, the processing time due to the use of autoregressive models, etc. is large. In addition, in entropy coding, the amount of calculation of the cumulative probability table increases exponentially with the number of dimensions. These are prospective solutions in the following ways; the spatial autoregressive module, which is mainly dominant in network processing, could be solved by introducing the parallel computation mechanism such as (He et al., 2021) and entropy coding by restricting the number of representative points and introducing cascade estimation such as (Zhu et al., 2022a).

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

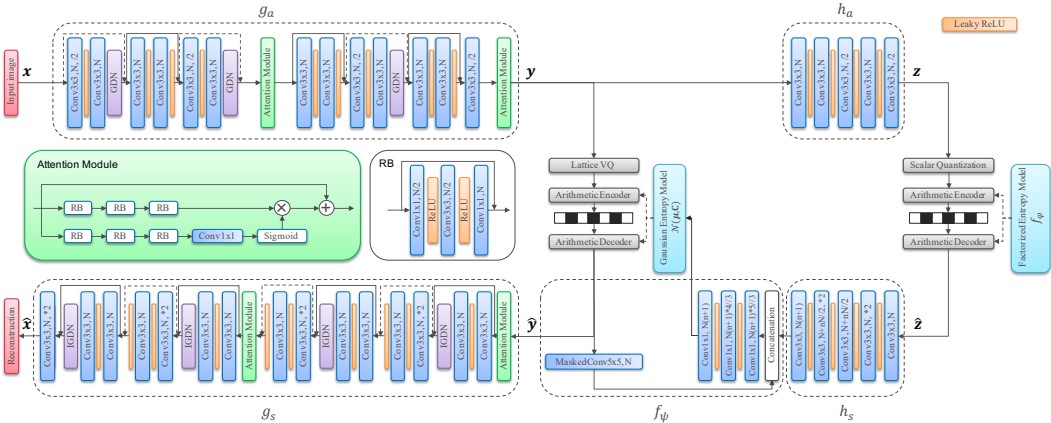

Figure 6: Network architecture

# A    IMPLEMENTATION DETAILS

## A.1    NETWORK ARCHITECTURE

Our network architecture is based on (Cheng et al., 2020) and is illustrated in Fig. 6. We use residual blocks and an attention module for both the feature encoder/decoder.

## A.2    LOW COMPLEXITY LATTICE VECTOR QUANTIZATION

We used fast quantization method (Conway & Sloane, 1982) for $A_2$, $D_4$, $E_8$ and (Conway & Sloane, 1986) for $\Lambda_{24}$. These methods calculate the Euclidean norm for a several candidate points (ex. 256 points for $\Lambda_{24}$) for each vector and selects the one with the smallest norm among them. These methods have two advantages: one is that it does not need to keep representative points in memory, and the other is that the quantization process can be performed at high speed without calculating the distance to all quantized representative points as in the conventional VQ method.

# B    ADDITIONAL EXPERIMENTS

We also tested on the CLIC2022 test set (CLIC, 2022) consisting of 30 high resolution images, and Tecnick image dataset (Asuni & Giachetti, 2014) consisting of 40 images with 1200 x 1200 resolutions.

RD performance results for CLIC2022 are shown in Fig. 7 and Tab. 3 and for Tecnick are shown in Fig. 8 and Tab. 4

For both CLIC2022 and Tecnick, the proposed method also gave the state-of-the-art performance. Especially for Tecnick, it showed relatively large performance gains compared to Kodak and CLIC2022. This may be due to the fact that Tecnick has a lower texture component compared to Kodak and CLIC2022. The proposed method tends to produce higher gains at lower rates than at higher rates, which may have contributed to Tecnick's higher performance.

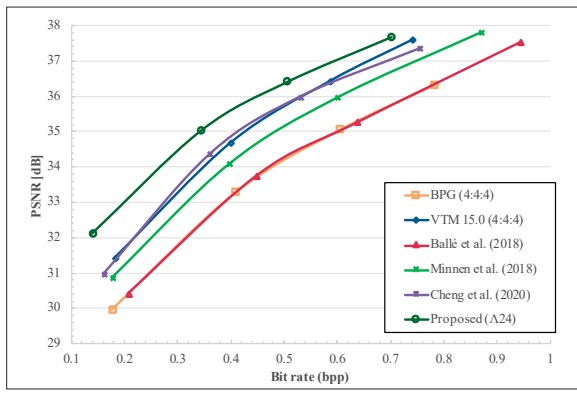

Table 3: Comparison of BDBR for CLIC2022 dataset

| Method | BDBR |
|---|---|
| BPG (4:4:4) | 40.9 % |
| VTM 15.0 (4:4:4) | 0.0 % (anchor) |
| Ballé et al. (2018) | 38.5 % |
| Minnen et al. (2018) | 13.5 % |
| Cheng et al. (2020) | -0.8 % |
| Proposed ($\Lambda_{24}$) | **-21.9 %** |

Figure 7: RD performance for CLIC2022 dataset

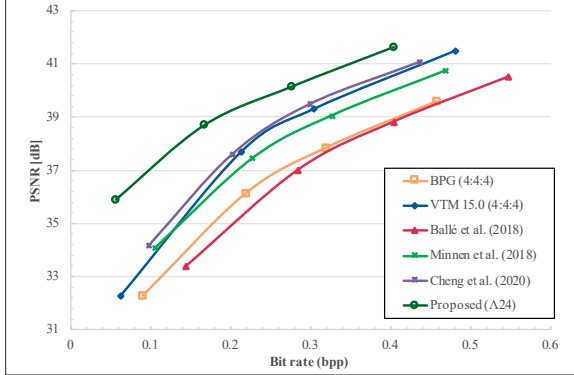

Table 4: Comparison of BDBR for Tecnick dataset

| Method | BDBR |
|---|---|
| BPG (4:4:4) | 45.8 % |
| VTM 15.0 (4:4:4) | 0.0 % (anchor) |
| Ballé et al. (2018) | 57.3 % |
| Minnen et al. (2018) | 13.1 % |
| Cheng et al. (2020) | -1.5 % |
| Proposed ($\Lambda_{24}$) | **-39.2 %** |

Figure 8: RD performance for Tecnick dataset

