# OpenReview forum: "LVQ-VAE:End-to-end Hyperprior-based Variational Image Compression with Lattice Vector Quantization"
_ICLR.cc/2023/Conference — Submitted to ICLR 2023_

### Official Review · Reviewer_9ZNU · 2022-10-24

**Confidence:** 5
**Correctness:** 3
**Technical Novelty And Significance:** 3
**Empirical Novelty And Significance:** 3
**Recommendation:** 3

**Clarity, Quality, Novelty And Reproducibility:**

The paper is easy to follow and is a serious work. The novelty is moderate. The paper provides enough details to reproduce the results.

**Strength And Weaknesses:**

Strengths:

(1) The idea of introducing Lattice Vector Quantization to learned image compression is interesting and novel.

(2) The RD performance of the proposed method looks impressive, showing up to 18% BD-rate saving over VTM.

Weaknesses:

(1) A similar work in CVPR2022 [1] that addresses Vector Quantization for learned image compression should be included for comparison and mentioned in related work. The novelty over [1] should be highlighted too.
[1] Zhu, Xiaosu, et al. "Unified Multivariate Gaussian Mixture for Efficient Neural Image Compression." Proceedings of the IEEE/CVF Conference on Computer Vision and Pattern Recognition. 2022.

(2) More ablation experiments should be conducted to analyze the complexity-performance trade-off among different choices of the lattice generating matrix. The complexity should include the model size, MAC, peak memory requirements, and encoding/decoding runtimes. It is unfortunate that these experiments are currently missing.

(3) In Eq. (8), the LVQ is modeled by additive uniform noise during training. This additive noise model does not appear to consider the generating matrix chosen for LVQ.

(4) The choice of the generating matrix is empirical and not well justified.


**Summary Of The Paper:**

This paper introduces Lattice Vector Quantization (LVQ) to hyperprior-based learned image compression. The LVQ is applied to co-located feature samples along the channel dimension. Moreover, it applies QMC, a traditional technique, to arrive at differentiable rate estimates. The LVQ is incorporated into Cheng’s learned codec (CVPR2020) to validate its coding performance.

**Summary Of The Review:**

The idea of introducing Lattice Vector Quantization (LVQ) to learned image compression is interesting and novel. Several well known LVQ techniques, including the design of the lattice generating matrix and the rate estimation, are extended to address learned compression. While it is intuitively agreeable that the coding performance may be improved further by introducing LVQ, the increased complexity is another issue that must be addressed. It is unfortunate that this aspect is currently missing. Moreover, one key reference is missing too.

---

> ### Author Response · Authors · 2022-11-18
> **Response to Reviewer 9ZNU**
>
> We would like to thank you for the review and suggestions.
>
> > (1) A similar work in CVPR2022 [1] that addresses Vector Quantization for learned image compression should be included for comparison and mentioned in related work. The novelty over [1] should be highlighted too. [1] Zhu, Xiaosu, et al. "Unified Multivariate Gaussian Mixture for Efficient Neural Image Compression." Proceedings of the IEEE/CVF Conference on Computer Vision and Pattern Recognition. 2022.
>
> We have added a comparison with the results of [1] and mentioned technical difference and novelty of the proposed method in the Introduction section. Specifically, the method [1] models the features assigned to representative points in a single distribution, whereas the proposed method models each feature in its own distribution, which has the advantage of removing redundancy among features.
>
> > (2) More ablation experiments should be conducted to analyze the complexity-performance trade-off among different choices of the lattice generating matrix. The complexity should include the model size, MAC, peak memory requirements, and encoding/decoding runtimes. It is unfortunate that these experiments are currently missing.
>
> Although we have not experimented with other lattices, the optimal lattices (A_2, D_4, E_8, \Lambda_24) for the uniform distribution are used in this paper. The uniform distribution is popularly adopted under high-rate assumption. In addition, the proposed model is expected to be optimized so that latent features approach the uniform distribution.
>
> For complexity, we understand that the complexity evaluation is important for practical deployment, but we give first priority to maximize coding efficiency and we believe this perspective is necessary in this research area. Therefore, the proposed method is not designed to fully take into account complexity reduction, and there is room for parameter optimization in terms of complexity reduction. At this moment, unfortunately, we cannot compare the proposed method with other methods in terms of encode/decode times, because the proposed method evaluates its R-D results by using the estimated results (-log(P)) instead of the practical coding results, in other words, the proposed method does not implement entropy encoder/decoder. We are planning to work on reducing complexity as a future work. In the Conclusion, we add our recognition that the processing time is a major issue at this moment, we have prospects for solving this issue by adopting parallel processing and restriction of representative points in entropy coder.
>
> > (3) In Eq. (8), the LVQ is modeled by additive uniform noise during training. This additive noise model does not appear to consider the generating matrix chosen for LVQ.
>
> u_i in equation (8) is a uniform noise depending on the lattice used. However, the symbol U used in section 3.1 was the same as in section 2, which may have caused confusion; the symbol U in section 3.1 was changed U to U_V0.
>
> > (4) The choice of the generating matrix is empirical and not well justified.
>
> We have added the reason why we used the lattices (A_2, D_4, E_8, \Lambda_24) in section 4.2. As mentioned above, we adopted these lattices based on the optimality for the uniform distribution. Therefore, the lattices used in this paper are considered theoretically appropriate.

---

### Official Review · Reviewer_fFaA · 2022-10-24

**Confidence:** 4
**Correctness:** 3
**Technical Novelty And Significance:** 3
**Empirical Novelty And Significance:** 2
**Recommendation:** 3

**Clarity, Quality, Novelty And Reproducibility:**

The clarity, quality and novelty of the work are ok, despite the reproducibility need to be improved and verified.

**Strength And Weaknesses:**

Strengths:
1.	The proposed entropy model is interesting and inspiring, i.e., approximating the discrete codeword probabilities by using the integration of continuous probability density.
2.	The overall performance on Kodak dataset is significant.

Weaknesses:
1.	Complexity and practicability. First of all, the authors didn’t provide any complexity evaluation, such as model size and encoding/decoding time. Despite the authors claimed that the use of lattice VQ solves the problem of codebook size bloating and provides fast quantization process, there is no experimental result to support this claim. More importantly, for lattice VQ, the complexity of practical entropy coding still exponentially increases with VQ dimensions. As mentioned in Section 3.2.2, the model calculates the cumulative probability table of a huge amount of sorted codewords, where the complexity is even larger than that of conventional vector quantization process. The entropy model needs to first produce and store the density integration of each codeword in the lattice VQ codebook, and it then obtains the cumulative probability table at both the encoder and decoder sides. In this paper, the VQ dimensions are set to 24, which is quite large for practical entropy coding. It is important to provide a detailed complexity evaluation for each part of the proposed method. Besides, are the claimed R-D results in Figure 4 & Figure 5 the practical coding results or the estimated results (by using -log(P))?
2.	More results on more datasets. The proposed method is only evaluated on Kodak with PSNR metric. What about the performance on high-resolution datasets such as CLIC-2021 and Tecnick? And it would be better to provide the results in terms of MS-SSIM.

Minor concerns:
1.	A related work [1] for VQ-based image compression is missing. Please provide the performance comparison and analysis.
[1] Zhu, Xiaosu, et al. "Unified Multivariate Gaussian Mixture for Efficient Neural Image Compression. CVPR2022


**Summary Of The Paper:**

The authors propose to incorporate lattice vector quantization into recent hyperprior-based image compression framework. Due to the nature of lattice VQ, the proposed method achieves better R-D performance without optimizing and storing a large VQ codebook. Furthermore, for end-to-end optimization, the codeword probabilities are approximated by the Monte Carlo integration of multivariate normal distribution. On Kodak dataset, the proposed method achieves 18% BD-rate saving over the latest video coding standard H.266/VVC.

**Summary Of The Review:**

Even though I think the idea is interesting and deserving of discussion (especially the entropy modeling), I still have concerns about the complexity and practicability of the proposed method, which play a crucial role for the task of image compression. And the experimental results are not enough to support the authors’ claims. I will reconsider based on the authors' response.

---

> ### Author Response · Authors · 2022-11-18
> **Response to Reviewer fFaA part1**
>
> We would like to thank you for the review and suggestions.
>
> > Weaknesses: 1. Complexity and practicability. First of all, the authors didn’t provide any complexity evaluation, such as model size and encoding/decoding time. Despite the authors claimed that the use of lattice VQ solves the problem of codebook size bloating and provides fast quantization process, there is no experimental result to support this claim. More importantly, for lattice VQ, the complexity of practical entropy coding still exponentially increases with VQ dimensions. As mentioned in Section 3.2.2, the model calculates the cumulative probability table of a huge amount of sorted codewords, where the complexity is even larger than that of conventional vector quantization process. The entropy model needs to first produce and store the density integration of each codeword in the lattice VQ codebook, and it then obtains the cumulative probability table at both the encoder and decoder sides. In this paper, the VQ dimensions are set to 24, which is quite large for practical entropy coding. It is important to provide a detailed complexity evaluation for each part of the proposed method. Besides, are the claimed R-D results in Figure 4 & Figure 5 the practical coding results or the estimated results (by using -log(P))?
>
> We understand that the complexity evaluation is important for practical deployment, but we give first priority to maximize coding efficiency and we believe this perspective is necessary in this research area. Therefore, the proposed method is not designed to fully take into account complexity reduction, and there is room for parameter optimization in terms of complexity reduction. At this moment, unfortunately, we cannot compare the proposed method with other methods in terms of encode/decode times, because the proposed method evaluates its R-D results by using the estimated results (-log(P)) instead of the practical coding results, in other words, the proposed method does not implement entropy encoder/decoder. We are planning to work on reducing complexity as a future work. In the Conclusion, we add our recognition that the processing time is a major issue at this moment, we have prospects for solving this issue by adopting parallel processing and restriction of representative points in entropy coder.
>
> As we mentioned above, the R-D results of the proposed method in Figure 4 & Figure 5 are the estimated results (-log2(P)), which has clearly noted in Section 4.2.
>
> > 2. More results on more datasets. The proposed method is only evaluated on Kodak with PSNR metric. What about the performance on high-resolution datasets such as CLIC-2021 and Tecnick? And it would be better to provide the results in terms of MS-SSIM.
>
> We have added a comparison experiment with the three latest methods in 2021 and 2022, including the method of Zhu et al. [1] below. The results show that our method outperforms all methods. Experimental results with CLIC2022 and Tecnick have also been added to Appendix B. The results show that the proposed method provides 21.9% gain for CLIC2022 and 39.2% gain for Tecnick compared to H.266/VVC. We didn't have the results for MS-SSIM at this moment, we are going to add the results for the final paper.

---

> ### Author Response · Authors · 2022-11-18
> **Response to Reviewer fFaA part2**
>
> > Minor concerns: 1. A related work [1] for VQ-based image compression is missing. Please provide the performance comparison and analysis. [1] Zhu, Xiaosu, et al. "Unified Multivariate Gaussian Mixture for Efficient Neural Image Compression. CVPR2022
>
> We have added a comparison with the results of [1] and technical difference and advantage of the proposed method are described in the Introduction section. Specifically, the method [1] models the features assigned to representative points in a single distribution, whereas the proposed method models each feature in its own distribution, which has the advantage of removing redundancy among features.
>
> > The clarity, quality and novelty of the work are ok, despite the reproducibility need to be improved and verified.
>
> We have added the LVQ method actually used in the experiment in the Appendix A.2. If you have any other questions about reproducibility, could you please describe them? We will add more details in final paper.
>
> > Even though I think the idea is interesting and deserving of discussion (especially the entropy modeling), I still have concerns about the complexity and practicability of the proposed method, which play a crucial role for the task of image compression. And the experimental results are not enough to support the authors’ claims. I will reconsider based on the authors' response.
>
> As you pointed out, naïve entropy coder suffers from high dimensional LVQ, because the number of "candidates" of representative points increases with the dimensionality. However, we believe that the following two ideas are useful to resolve this problem. One idea is to limit the number of representative points as described above. Another idea is to eliminate the computation of cumulative probability for unused representative points. Although entropy coder/decoder accumulate the probability of each codeword (corresponding to each representative point), it is not necessary to compute cumulative probability for all candidates of representative point. Since the proposed method can estimate the parameters of the probability distribution with high precision, it is expected that the number of representative points to be calculated can be reduced by calculating them in descending order of probability (in the proposed method, in order of closest to the mean of the estimated probability distribution).

---

### Official Review · Reviewer_nBnd · 2022-10-24

**Confidence:** 4
**Correctness:** 4
**Technical Novelty And Significance:** 3
**Empirical Novelty And Significance:** 3
**Recommendation:** 5

**Clarity, Quality, Novelty And Reproducibility:**

This paper is overall of good quality and some novelty. The layout of figures and tables can be improved. Some figs and tables are not well aligned. (For example: (Fig4 and Table 1), (Fig5 and Table2)). A related work [1] should be included for discussion and comparison.
[1] Zhu, Xiaosu, et al. "Unified Multivariate Gaussian Mixture for Efficient Neural Image Compression. CVPR2022



**Strength And Weaknesses:**

Experiments including the ablation study are convincing. The performance is also good.
A more detailed introduction about some aspects of lattice vector quantization is preferred. For example, how the vector quantization matrix generation is performed during training.
For the experiment part, the effect of this quantization method on the computational complexity and memory consumption during training and inference is also expected. It will also be better if more experiments on different anchors (except Cheng) and different test datasets are performed.


**Summary Of The Paper:**

This paper introduced Lattice Vector Quantization into VAE-based image compression framework. Besides, a multivariate normal distribution including covariance matrix parameters are proposed. Experiments demonstratedd the effectiveness of these two features with a 18% BD-rate save compared with VTM15.0.


**Summary Of The Review:**

This paper demonstrated that Lattice Vector Quantization can further improve the coding performance of VAE-based image compression framework. Experiments show the effectiveness but the conclusion can be stronger if more experiments can be performed. If the authors can provide more information about the implementation details of lvq, we can give a more solid recommendation.

---

> ### Author Response · Authors · 2022-11-18
> **Response to Reviewer nBnd**
>
> We would like to thank you for the review and suggestions.
>
> > Experiments including the ablation study are convincing. The performance is also good. A more detailed introduction about some aspects of lattice vector quantization is preferred. For example, how the vector quantization matrix generation is performed during training. For the experiment part, the effect of this quantization method on the computational complexity and memory consumption during training and inference is also expected.
>
> LVQ does not perform matrix generation; the generating matrix is uniquely defined by the type of lattice. To make this clear, we have specified this in section 3.1. In addition, a fast LVQ method is used in this experiment. This fast quantization method does not need to store representative points and offers low complexity. This is clearly stated in Appendix A.2.
>
> > It will also be better if more experiments on different anchors (except Cheng) and different test datasets are performed.
>
> We have added a comparison experiment with the three latest methods in 2021 and 2022, including the method of Zhu et al. [1] below. The results show that our method outperforms all methods. Experimental results with CLIC2022 and Tecnick have also been added to Appendix B. The results show that the proposed method provides 21.9% gain for CLIC2022 and 39.2% gain for Tecnick compared to H.266/VVC.
>
> > This paper is overall of good quality and some novelty. The layout of figures and tables can be improved. Some figs and tables are not well aligned. (For example: (Fig4 and Table 1), (Fig5 and Table2)).
>
> (Fig.4 and Tab.1) and (Fig.5 and Tab.2) were changed to top-aligned, respectively.
>
> > A related work [1] should be included for discussion and comparison. [1] Zhu, Xiaosu, et al. "Unified Multivariate Gaussian Mixture for Efficient Neural Image Compression. CVPR2022
>
> We have added a comparison with the results of Zhu et al. [1]. In addition, we have described the differences and advantages of the proposed method in the Introduction section. In particular, Zhu et al.'s method models the features assigned to representative points in a single distribution, whereas the proposed method models each feature in its own distribution, which has the advantage of removing redundancy among features.

---

### Official Review · Reviewer_SVp9 · 2022-10-24

**Confidence:** 3
**Correctness:** 4
**Technical Novelty And Significance:** 2
**Empirical Novelty And Significance:** 3
**Recommendation:** 5

**Clarity, Quality, Novelty And Reproducibility:**

The overall quality of the paper is quite good.

Novelty is medium. While the results are SOTA and I agree with the authors' claim that previous work has not used LVQ with a hyperprior and end-to-end optimization, all of the components do already exist.

Reproducibility seems fairly difficult so releasing code to set up the lattice and approximate the likelihoods would help significantly (the other components already exist in opensource projects like TensorFlow-Compression and CompressAI).


**Strength And Weaknesses:**

The primary strength of this paper is the achieving SOTA rate-distortion performance. As far as I know, now other papers or standard codecs achieve as good a compression rate when using PSNR as the quality metric.

The primary weakness of the paper is a lack of runtime data. I suspect this method is extremely slow since spatially AR context models are known to be slow due to the difficulty of parallelizing the decoding process, and the prediction of a full covariance matrix leads to a large number of output parameters (if I understand the setup correctly, 192 channels using \Lambda_24 leads to 8 groups of 24D vectors and thus 8 * (24^2 + 24) = 4800 parameters per spatial location in the latent image representation). I suspect this leads to decode times of multiple seconds, even on fast hardware, when a deployable model needs to decode kodak-size images in a handful of milliseconds. To the best of my knowledge, no neural methods achieve decode speeds on par with standard codecs like VTM (or BPG or JPEG, etc.), but other models are probably two orders of magnitude faster than the method in this paper while only losing 5-10% in terms of rate.

The other weakness is a lack of direct comparison against stronger neural methods. Balle 2018, Minnen 2018, and Cheng 2020 are all relatively old methods. In particular, part of the motivation of this paper is the lack of modeling the dependencies between channels (SQ doesn't do this, VQ does). But existing papers adopt other methods for (partially) modeling these dependencies, for example:

Channel-wise Autoregressive Entropy Models for Learned Image Compression
David Minnen, Saurabh Singh
(ICIP 2020) https://arxiv.org/abs/2007.08739

ELIC: Efficient Learned Image Compression with Unevenly Grouped Space-Channel Contextual Adaptive Coding
Dailan He, Ziming Yang, Weikun Peng, Rui Ma, Hongwei Qin, Yan Wang
(CVPR 2022) https://arxiv.org/abs/2203.10886

Transformer-based Transform Coding
Yinhao Zhu, Yang Yang, Taco Cohen
(ICLR 2022) https://openreview.net/forum?id=IDwN6xjHnK8

The channel-wise AR approach used by these papers should be mentioned and compared to the LVQ approach adopted here since they both address the same underlying deficiency. I believe the RD performance of the LVQ approach is better than what is reported in these papers. However, the runtime is vastly different (ELIC in particular is focused on balancing RD performance with runtime), and the preferable trade-off probably falls to ELIC over the proposed LVQ method, i.e. I wouldn't accept 10-100x slower decoding to save 5-10% on file sizes.

Finally, I'd like to better understand how much of the RD performance is due to the LVQ vs. the full covariance matrix. Fig 4 and Table 1 partially answer the question by showing that for several lattice dimensionalities, predicting a full covariance matrix helps (as expected). Another interesting comparison would be to use a trivial lattice (hyper-cubes) but still group the channels into 24D vectors and predict a full covariance matrix. I expect \Lambda_24 is better (the underlying theory predicts this for mse) but the size of the gap would make it clear whether or not the added complexity of \Lambda_24 is warranted.


**Summary Of The Paper:**

This paper presents a neural image compression models based on lattice vector quantization (LVQ).

The authors note two deficiencies in existing work: (1) most models use scalar quantization (SQ), which is typically less powerful than vector quantization (VQ) if a large enough codebook is used, and (2) models that do use VQ do not use a hyperprior or end-to-end optimization. The authors address these problems by using LVQ, which has a large, though constrained, codebook, and they show how to jointly optimize a distribution over the lattice (the entropy model) with a hyperprior and spatially autoregressive model.

Two other important contributions are the evaluation of predicting a full covariance matrix for each LVQ group, rather than a diagonal covariance matrix, and using Monte Carlo integration to more quickly calculate (an approximation of) the likelihood of each feature vector.

The result is, as far as I know, state of the art rate-distortion (RD) performance for lossy image compression. The evaluation shows an 18% rate savings over VTM, one of the best standard (hand-engineered) image codecs, as well as gains over other learning-based (neural) codecs (Fig 5 and Table 2). The paper also compares different lattice dimensionalities and the effect of predicting a full covariance matrix (Fig 4 and Table 1).


**Summary Of The Review:**

As far as I know, this paper presents state-of-the-art rate-distortion results for lossy image compression. That's a significant achievement given the existing work on this problem.

That said, the improvement in rate-savings is only moderate compared to the best existing methods (see references above), and the other methods appear to support much faster decode speeds, which is crucial for deployment.

So my recommendation is "5: marginally below the acceptance threshold" since it's not entirely clear how much of the performance gain comes from LVQ vs. predicting a full covariance matrix within each VQ group vs. the spatial autoregressive model.

---

> ### Author Response · Authors · 2022-11-18
> **Response to Reviewer SVp9 part1**
>
> We would like to thank you for the review and suggestions.
>
> > The primary weakness of the paper is a lack of runtime data. I suspect this method is extremely slow since spatially AR context models are known to be slow due to the difficulty of parallelizing the decoding process, and the prediction of a full covariance matrix leads to a large number of output parameters (if I understand the setup correctly, 192 channels using \Lambda_24 leads to 8 groups of 24D vectors and thus 8 * (24^2 + 24) = 4800 parameters per spatial location in the latent image representation). I suspect this leads to decode times of multiple seconds, even on fast hardware, when a deployable model needs to decode kodak-size images in a handful of milliseconds. To the best of my knowledge, no neural methods achieve decode speeds on par with standard codecs like VTM (or BPG or JPEG, etc.), but other models are probably two orders of magnitude faster than the method in this paper while only losing 5-10% in terms of rate.
>
> We understand that the complexity evaluation is important for practical deployment, but we give first priority to maximize coding efficiency and we believe this perspective is necessary in this research area. Therefore, the proposed method is not designed to fully take into account complexity reduction, and there is room for parameter optimization in terms of complexity reduction. At this moment, unfortunately, we cannot compare the proposed method with other methods in terms of encode/decode times, because the proposed method evaluates its R-D results by using the estimated results (-log(P)) instead of the practical coding results, in other words, the proposed method does not implement entropy encoder/decoder. We are planning to work on reducing complexity as a future work. In the Conclusion, we add our recognition that the processing time is a major issue at this moment, we have prospects for solving this issue by adopting parallel processing and restriction of representative points in entropy coder.
>
>
>
> > The other weakness is a lack of direct comparison against stronger neural methods. Balle 2018, Minnen 2018, and Cheng 2020 are all relatively old methods. In particular, part of the motivation of this paper is the lack of modeling the dependencies between channels (SQ doesn't do this, VQ does). But existing papers adopt other methods for (partially) modeling these dependencies, for example:
> >
> > Channel-wise Autoregressive Entropy Models for Learned Image Compression David Minnen, Saurabh Singh (ICIP 2020) https://arxiv.org/abs/2007.08739
> >
> > ELIC: Efficient Learned Image Compression with Unevenly Grouped Space-Channel Contextual Adaptive Coding Dailan He, Ziming Yang, Weikun Peng, Rui Ma, Hongwei Qin, Yan Wang (CVPR 2022) https://arxiv.org/abs/2203.10886
> >
> > Transformer-based Transform Coding Yinhao Zhu, Yang Yang, Taco Cohen (ICLR 2022) https://openreview.net/forum?id=IDwN6xjHnK8
>
> > The channel-wise AR approach used by these papers should be mentioned and compared to the LVQ approach adopted here since they both address the same underlying deficiency. I believe the RD performance of the LVQ approach is better than what is reported in these papers. However, the runtime is vastly different (ELIC in particular is focused on balancing RD performance with runtime), and the preferable trade-off probably falls to ELIC over the proposed LVQ method, i.e. I wouldn't accept 10-100x slower decoding to save 5-10% on file sizes.
>
> We have added descriptions of the above three papers in the Introduction section and described the difference between these methods and our method for channel direction de-correlation in section 3.1. We also added the results of the latest methods in 2021 and 2022, including ELIC mentioned above, in the Experimental section. As mentioned above, there is no comparison on complexity, but we found that for Tecnick (which have added updated version) the proposed method provides significant gain of 39.2% compared to H.266/VVC and outperforms the other learning-based methods, and we believe that it can bring a new perspective to the readers.
>
> > Finally, I'd like to better understand how much of the RD performance is due to the LVQ vs. the full covariance matrix. Fig 4 and Table 1 partially answer the question by showing that for several lattice dimensionalities, predicting a full covariance matrix helps (as expected).
>
> At this moment, there are no results for the full covariance matrix. However, since the LVQ and the full covariance matrix are independent of each other, the gain obtained by the full covariance matrix only should be the difference between the use and non-use of a covariance matrix in Table 1. We will add the results on the full covariance matrix in time for the final paper.

---

> ### Author Response · Authors · 2022-11-18
> **Resposen to Reviewer SVp9 part2**
>
> > Another interesting comparison would be to use a trivial lattice (hyper-cubes) but still group the channels into 24D vectors and predict a full covariance matrix. I expect \Lambda_24 is better (the underlying theory predicts this for mse) but the size of the gap would make it clear whether or not the added complexity of \Lambda_24 is warranted.
>
> Since trivial lattice (hyper-cubes) is the same as scalar quantization, it does not improve performance. For the use of \Lambda_24, we applied fast quantization method (added the details in Appendix A.2), which have little effect on complexity.
>
> > Reproducibility seems fairly difficult so releasing code to set up the lattice and approximate the likelihoods would help significantly (the other components already exist in opensource projects like TensorFlow-Compression and CompressAI).
>
> We do not plan to release the source code at this moment, but we have stated in the Appendix A.2 that we used a fast method for the LVQ implementation. If you have any other questions about reproducibility, could you please describe them? We will add more details in final paper.

---

### Author Response · Authors · 2022-11-18
**Revised paper uploaded**

We would like to express our thanks to all the reviewers for their valuable feedback and suggestions. We have uploaded a revised version of the paper.
The core changes are summarized as follows:
1. We have conducted additional experiments: new comparisons with the three most recent methods for 2021, 2022, and added results for the CLIC2022 and Tecnick data sets. The results show that our method outperforms all methods and provides 21.9% gain for CLIC2022 and 39.2% gain for Tecnick compared to H.266/VVC.
2. We have added descriptions of several related studies and added differences and novelties from the proposed method in Introduction and section 3.1.
3. We have added the reason why we used the lattices (A_2, D_4, E_8, \Lambda_24) in section 4.2. The reason is that these lattices are optimal for a uniform distribution and the proposed model is expected to be optimized so that latent features approach the uniform distribution.
4. We have added details on the LVQ implementation to the Appendix A.2.
5. We have stated that the processing time is a major issue at this moment, but we have prospects for solving this issue in Conclusion.

---

### Decision · Program_Chairs · 2023-01-20

**Decision:**

Reject

**Justification For Why Not Higher Score:**

While I am not an expert in the domain of image compression, all reviewers have provided in-depth comments about the paper and they share similar concerns. (I believe Reviewer SVp9 is one of the leading experts in this field and the submission cited many of his papers.) The reviewers questioned the runtime complexity of the proposed method, which the authors could not provide at this point. Thus I am confident that the paper is not ready to be published in its current form.

**Justification For Why Not Lower Score:**

N/A

**Metareview: Summary, Strengths And Weaknesses:**

This paper presents a neural image compression models based on lattice vector quantization (LVQ). Instead of scalar quantization (SQ), the authors use the more powerful than vector quantization (VQ) together with a hyperprior for end-to-end optimization.

Strength:

The authors achieve state of the art rate-distortion (RD) performance for lossy image compression. The paper also compares different lattice dimensionalities and demonstrate the effect of predicting a full covariance matrix.

Weakness:

All reviewers raised the concern about the runtime complexity of the proposed method. They suspect the proposed method is too slow to be practical.